# Flexible Strain Sensors Based on Thermoplastic Polyurethane Fabricated by Electrospinning: A Review

**DOI:** 10.3390/s24154793

**Published:** 2024-07-24

**Authors:** Zhiyuan Zhou, Weirui Tang, Teer Xu, Wuyang Zhao, Jingjing Zhang, Chuanwu Bai

**Affiliations:** School of Tropical Agriculture and Forestry, Hainan University, Danzhou 571799, China; 20213006647@hainanu.edu.cn (Z.Z.); 20213005555@hainanu.edu.cn (W.T.); 20213005572@hainanu.edu.cn (T.X.); 20213005554@hainanu.edu.cn (W.Z.)

**Keywords:** flexible strain sensors, electrospinning, thermoplastic polyurethane, conductive materials, applications

## Abstract

Over recent years, thermoplastic polyurethane (TPU) has been widely used as a substrate material for flexible strain sensors due to its remarkable mechanical flexibility and the ease of combining various conductive materials by electrospinning. Many research advances have been made in the preparation of flexible strain sensors with better ductility, higher sensitivity, and wider sensing range by using TPU in combination with various conductive materials through electrospinning. However, there is a lack of reviews that provide a systematic and comprehensive summary and outlook of recent research advances in this area. In this review paper, the working principles of strain sensors and electrospinning technology are initially described. Subsequently, recent advances in strain sensors based on electrospun TPU are tracked and discussed, with a focus on the incorporation of various conductive fillers such as carbonaceous materials, MXene, metallic materials, and conductive polymers. Moreover, the wide range of applications of electrospun TPU flexible strain sensors is thoroughly discussed. Finally, the future prospects and challenges of electrospun TPU flexible strain sensors in various fields are pointed out.

## 1. Introduction

With the rapid advances in the Internet of Things (IoT), Industry 4.0, big data, artificial intelligence (AI), robotics, and digital health, there is an increasing demand for high-precision sensing at all places and times [1]. Strain sensors can convert various types of deformations into changes in electrical signals to detect the dynamic characteristics of physical states, which have garnered widespread attention from many scholars [2]. However, traditional strain sensors made from rigid materials such as metals, ceramics, and semiconductors face inherent performance limitations such as lack of flexibility and ductility, making it difficult to meet growing demands [3]. As shown in Figure 1, flexible strain sensors have been studied and applied in many fields of intelligent electronic devices such as intelligent electronic skin [4], human–computer interaction [5], sports performance monitoring [6], and flexible robotic [7] due to their excellent properties of flexibility, stretchability, breathability, high sensitivity, and strong stability. Flexible strain sensors can be trimmed to different sizes and even folded into different shapes for a variety of complex and changing applications, which greatly broadens their application areas [8].

The performance of flexible strain sensors mainly depends on the mechanical properties of the flexible polymer matrix and the electrical properties of conductive fillers [10]. Common materials used as flexible polymer matrices include thermoplastic polyurethane (TPU) [11], polyimide (PI) [12], polyetheretherketone (PEEK) [13], polyethersulfone (PES) [14], polycarbonate (PC) [15], polyethylene naphthalate (PEN) [16], hydrogel [17], and polyethylene terephthalate (PET) [18]. Among these, TPU exhibits superior overall performance. It not only has excellent abrasion resistance, exceeding that of natural rubber by a factor of five (far superior to PI) but also boasts high transparency—in contrast to PEEK, which is usually opaque or translucent [19,20]. TPU’s strong flexibility and mechanical elasticity surpass those of PES, and its economic benefits over hydrogel are realized at a lower cost [21,22]. In terms of stability, TPU’s chemical resistance is better than PET’s, and it has excellent thermal stability and adaptability to low-temperature environments compared to PC [23,24]. Additionally, TPU is easy to process and can be recycled, making it more environmentally friendly than PEN [16]. These combined advantages make TPU one of the most widely used flexible polymer matrices for flexible strain sensors in practical applications. However, pure TPU is essentially a non-conductive material and cannot be directly used to fabricate sensing elements [25]. Electrospinning technology can further process TPU nanofibers in combination with various conductive fillers to endow the composite TPU nanofibers with high electrical conductivity, enabling the preparation of high-performance flexible strain sensors.

Currently, researchers have explored the use of carbon materials (e.g., carbon black, carbon nanotubes, and graphene) [26,27,28], metallic nanomaterials (e.g., nanoparticles, nanorods, and nanowires) [29,30,31], and conductive polymers (e.g., PPy and PANI) [32,33]. They have prepared numerous flexible strain sensors with excellent performance by combining these materials with electrospun TPU nanofiber substrates. However, there is a lack of review papers that provide a systematic and comprehensive summary and outlook of these recent research results. For example, Backes et al. explored TPU synthesis, fabrication techniques, blends, composites, and applications, but did not focus on the most advanced TPU prepared by electrospinning techniques and their application in flexible sensors [34]. Chen et al. introduced the research progress of flexible strain sensors based on conductive fillers and TPU but did not analyze the most fruitful electrospun TPU flexible strain sensors, and their review was published three years ago, which affects its novelty and timeliness [11]. In this review, Section 1 provides a general introduction to the working principles of strain sensors and electrospinning technology, highlighting the salient features of electrospun TPU strain sensors. Section 2 summarizes recent research advances in various conductive fillers for electrospun TPU flexible strain sensors, with a focus on carbonaceous materials, MXene, metallic materials, and conductive polymers. Section 3 offers a comprehensive discussion of various sensing applications of electrospun TPU flexible strain sensors, including motion detection, health monitoring, recognition of facial expressions and speech, electronic skin, and multifunctional electrospun TPU sensors. Lastly, the future research trends and challenges of electrospun TPU flexible strain sensors are proposed. This review comprehensively and systematically summarizes the recent advances in electrospun TPU-based flexible strain sensors, bridges the gap in the field, and serves as a unique guideline for subsequent research on electrospun TPU-based flexible strain sensors.

## 2. Electrospinning TPU Strain Sensors

### 2.1. Strain Sensors

Strain sensors can convert various external physical stimuli (e.g., stretching, compression, twisting, etc.) into recognizable electrical signal outputs. According to the transduction mechanism, strain sensors can be classified into resistive [35], capacitive [36], voltage (piezoelectric and triboelectric) [37,38], and inductance/magnetism types [39,40]. Resistive strain sensors are widely used due to their significant advantages, such as simple preparation processes, easy signal reception, and stable signal output [41]. Their mode of operation involves using the resistance strain gauge to quantify the response to the direction and degree of strain applied to the sensor through changes in the piezoresistance ratio or increases in the inherent resistance of the material when the conductive layer is deformed [42]. Initially, resistive strain sensors were used to detect the fatigue level of materials. However, with in-depth research on composite nanomaterials, flexible resistive strain sensors based on conductive polymer composites (CPCs) are now increasingly applied in high-end fields, such as intelligent wearable devices and electronic skin [43]. 

The working principle of CPC strain sensors is mainly based on the tunneling effect, percolation theory, and crack propagation. According to the electron tunneling effect, the conductive network is generated by the high-speed movement of electrons in the conductive pathway based on conductive filler loading [44,45]. There is a potential energy barrier between adjacent but unconnected conductive fillers [46]. Under applied stress, the spacing between adjacent conductive fillers changes, and according to quantum mechanics, electrons may pass through the potential barrier between adjacent conductive fillers, resulting in the rearrangement of the conductive network, which is converted into a macroscopic electrical signal output. Percolation theory suggests that when the conductive filler in the composite system is increased to a certain critical amount, the electrical resistance decreases sharply [47]. Meanwhile, the resistivity-conductive filler dosage curve shows a narrow region of abrupt change, where a small change in the amount of conductive filler leads to a significant change in resistivity [48]. The conductivity of CPC strain sensors is described using percolation theory based on the tunneling effect, with the following equation:(1)$$\sigma \propto {\left(\emptyset -\emptyset_{c}\right)}^{t},\emptyset >\emptyset_{c}$$
where σ is the electrical conductivity of the composites, ∅ is the mass fraction of conductive filler, $\emptyset_{c}$ represents the percolation threshold, and t is a critical exponent in relation to the dimensionality of the conductive network. Crack propagation means that when the conductive filler is stretched to a certain degree, cracks will appear on the surface or inside the conductive filler, which will lead to a change in the contact area and thus cause a change in resistance [49]. This will be specifically analyzed in the next section.

GF is introduced to evaluate the response sensitivity of the strain sensor, defined as the ratio of the relative change in resistance to the mechanical strain. GF is analyzed using the slope of the change curve, which can be calculated from the following equation:(2)$$GF=\frac{\Delta R/R_{0}}{\varepsilon }=\frac{\left(R_{t}-R_{0}\right)/R_{0}}{\varepsilon },\varepsilon =\Delta L/L_{0}$$
where $R_{0}$ is the initial resistance, $R_{t}$ is the test resistance, and ε is the applied strain.

### 2.2. Electrospinning Techniques

Electrospinning technology is a process in which a polymer solution undergoes jet stretching under the action of high-voltage electrostatic field forces, and through solvent evaporation, continuous, ultrafine fibers, as well as micron-sized, ultra-thin fiber membranes, are prepared [50,51,52]. The basic device consists of four main components: a high-voltage power supply that provides high-voltage electrostatic field force, a syringe with a jet needle, a syringe pump, and a grounded receiving device [53]. In the process of electrospinning, the liquid extruded from the nozzle forms hanging droplets under the action of surface tension, while the high-voltage electrostatic field produces a transient potential difference between the needle and the receiver, causing the extruded droplets to take on a charge, with the repulsive force between the surface charges of the droplets counteracting the surface tension [54]. As the voltage increases, the repulsive surface charge force dominates and the hemispherical liquid is elongated, deforming the droplet into a stable Taylor cone [55]. The charged fluid is ejected from the Taylor cone under the action of an electrostatic field and stretched within the electrostatic field [56]. Initially, the charged fluid undergoes linear stretching. Under the action of the electrostatic field, electrical bending instability (whipping instability) grows [57]. The ejected fluid thins under stretching and solidifies rapidly until solid fibers are deposited on the collector [58]. By controlling the parameters of electrostatic spinning, nanofibers with different morphologies can be prepared, such as uniaxially and biaxially arranged nanofiber meshes, ribbons, porous fibers, necklace-like fibers, nanowebs, hollow fibers, nanowire-in-microtube, and multichannel tubules [51] (Figure 2).

Electrospinning technology provides a unique and effective way to prepare TPU nanofiber materials. It has advantages in microstructure control and can produce nanofibers with a high specific surface area and fine pore structure [50]. Compared to other preparation processes (e.g., melt mixing, solution mixing, solvent casting, particle leaching, thermally induced phase separation (TIPS), 3D printing, and injection molding), electrospinning exhibits unique benefits: it offers precise control over microstructure, unlike melt blending; it reduces the use of hazardous organic solvents, showing better environmental compatibility than solution blending; and it simplifies the production process and shortens preparation time compared to TIPS [59]. Although 3D printing technology can produce complex geometries, electrospinning excels in microstructural consistency and orientation [60]. Injection molding, while suitable for rapid production, is less precise than electrospinning in microstructure control [61].

### 2.3. Characterized Electrospun TPU Strain Sensors

Table 1 demonstrates the effect of different electrospinning process parameters on the morphological properties of TPU [61]. Electrospinning technology endows TPU nanofibers with high porosity, high uniformity, and high-quality microstructure. The prepared electrospun TPU nanofibers have a large specific surface area, which is conducive to the bonding and loading of more conductive fillers [62]. Therefore, the most common and effective method to prepare electrospun TPU strain sensors is to use electrospun TPU fibers as a substrate and introduce conductive fillers. This method combines the high tensile properties of TPU nanofibers with the high electrical conductivity of conductive fillers. Highly conductive nanoparticles form continuous conductive pathways, which are carried and protected by a flexible matrix made of TPU nanofibers with high stretch-recovery properties [63].

## 3. The Use of Conductive Fillers in Electrospun TPU Strain Sensors

Among all types of conductive fillers applied in electrospun TPU sensors, carbonaceous materials, MXene, metallic materials, and conductive polymers are the most common. In this section, recent research advances in various conductive fillers for electrospun TPU flexible strain sensors are discussed, especially on carbonaceous materials, MXene, metallic materials, and conductive polymers. Details of some typical electrospun TPU strain sensors are summarized in Table 2.

### 3.1. Carbonaceous Materials

#### 3.1.1. Zero-Dimensional Carbonaceous Material—CB

Zero-dimensional carbon black (CB) possesses outstanding advantages, including good dispersion in solvents, high conductivity, and small size, which make it an ideal conductive nanofiller [93]. The high specific surface area and low dimensionality of the conductive CB particles facilitate proper dispersion in the electrospun TPU substrate and establish stable interactions with the thermoplastic polyurethane chains, endowing the composites with excellent electromechanical properties [94]. However, the point-to-point mobile conductive network constructed by conductive CB particles in thermoplastic polyurethanes is easily deformed, which makes the composites exhibit high conductive sensitivity under tensile strain [95]. Conductive CB particles are widely used as conductive fillers in electrospun TPU strain sensors due to the ease of processing and economic practicality for large-scale industrial production [96]. 

A common strategy is to prepare CB/TPU composite nanofiber membranes by embedding CB particles in TPU nanofiber membranes using the ultrasonication method to construct the conductive network [64] (Figure 3a). The CB/TPU flexible strain sensor prepared from this composite nanofiber membrane exhibits a combination of high sensitivity under tensile strain (GF of 8962.7 at 155% strain), fast response time (60 ms), excellent stability and durability (greater than 10,000 cycles), and a wide workable tensile range (0–160%). On this basis, some researchers have prepared CB/TPU/Ecoflex strain sensors using Ecoflex to encapsulate ultrasonically modified CB/TPU composite nanofibers to form a sandwich structure [65] (Figure 3b). This sandwich structure provides effective protection for the conductive CB/TPU fiber network, giving the strain sensor excellent sensing performance, including a low detection limit (0.5% strain), a wide response range (up to 225% strain), ultra-high sensitivity (maximum GF of 3186.4 at 225% strain), fast response time (70 ms), and good repeatability (over 5000 tension/release cycles). In addition, the strain sensor has excellent immunity to external humidity and temperature.

Nevertheless, the application of conductive CB particles in electrospun TPU strain sensors faces certain limitations. Due to their zero-dimensional shape, CB particles are prone to aggregation in the electrospun TPU matrix, which inevitably results in an increase in conductivity percolation and affects the overall conductivity of the strain sensor [97]. Additionally, the interactions between CB particles are extremely weak, which makes the constructed point-to-point mobile conductive network easily susceptible to destruction under strain, affecting the overall tensile performance of the strain sensor [98]. 

The section discusses the use of zero-dimensional CB as a conductive nanofiller in electrospun thermoplastic polyurethane (TPU) strain sensors. CB is valued for its dispersion in solvents, high conductivity, and small size. It forms a point-to-point conductive network in TPU, enhancing electromechanical properties and sensitivity to tensile strain. However, CB’s tendency to aggregate can increase conductivity percolation and affect sensor conductivity. The section also highlights the use of CB/TPU composite nanofiber membranes and the creation of sandwich structures for improved sensor performance, including sensitivity, response time, and durability.

#### 3.1.2. One-Dimensional Carbonaceous Material—CNTs

Among carbonaceous materials, carbon nanotubes (CNTs) are considered the most suitable conductive fillers for application in electrospun TPU flexible sensors. Compared to CB, CNTs have a larger aspect ratio, which results in extremely low conductivity percolation in CNT/TPU conductive composites [99]. Additionally, CNTs possess a typical hexagonal and linear entangled structure that is perfectly connected through chemical bonds. This allows them to form conductive pathways and construct conductive networks in electrospun TPU substrates even at low concentrations [100]. CNTs are categorized into single-walled carbon nanotubes (SWCNTs) and multi-walled carbon nanotubes (MWCNTs) based on the number of graphene layers within the carbon nanotube structure. Ultra-high aspect ratio CNTs are widely used in electrospun TPU strain sensors because they create an end-to-end mobile conductive network in the electrospun TPU substrate, providing the sensors with excellent electrical conductivity [101].

The conductivity of electrospun TPU strain sensors depends on the content and dispersion state of the conductive filler within the electrospun TPU matrix [102]. When the CNT content in the electrospun TPU substrate is appropriately increased, the distance between nanotubes decreases, forming a more stable end-to-end mobile conductive network, thus improving the sensor’s overall conductivity [103]. Therefore, one way to prepare highly conductive electrospun TPU strain sensors is to appropriately increase the CNT content within the electrospun TPU matrix. Some researchers have modified TPU nanofiber membranes by rapidly synthesizing dopamine (DA) to form a polydopamine (PDA) coating on the surface, significantly improving the membranes’ ability to load CNTs [66,104]. This is primarily due to DA undergoing oxidative self-polymerization with oxygen in humid air in a weakly alkaline environment, forming oligomers that continuously cross-link into polymers with higher molecular weights. DA, its oxidation products, and its polymers self-assemble in solution through various covalent and non-covalent bonds, forming the PDA layer. PDA has extremely strong adhesion ability and can enhance the deposition fastness of CNTs on the TPU nanofiber membrane surface. Meanwhile, the abundant hydroxyl and amino groups in PDA molecules increase the hydrophilicity of the TPU nanofiber membrane. This facilitates more CNTs in the suspension to penetrate the pores of the TPU nanofiber membrane, thus improving deposition efficiency. Flexible strain sensors prepared based on the CNT/DA/TPU composite nanofiber membrane exhibit excellent sensitivity (GF of 10,528.53 at 200% strain), fast response/recovery time (188/211 ms), wide sensing range (up to 200%), and excellent stability and durability [67] (Figure 4a). On this basis, some researchers explored constructing hydrophobic layers consisting of polyhedral oligomeric silsesquioxane (POSS), and 1H,1H,2H,2H-perfluorooctyltrimethoxysilane (FAS) on CNT/DA/TPU composite nanofiber mats. POSS as a protective layer increased the sensing range of the flexible sensor to 550%, while the lower surface energy of the -CF2- and -CF3 groups in FAS made the sensor superhydrophobic [68] (Figure 4b). 

However, the tensile properties of the composites are negatively affected by increasing the content of CNTs in the TPU matrix [105]. To improve the electrical conductivity of TPU composites while maintaining their tensile properties, methods such as controlling the content of carbon nanotubes and lowering the percolation threshold have been explored. Several researchers have fabricated a high-performance CNT/TPU@SBS flexible strain sensor by constructing a double-percolation structure consisting of a continuous CNT/TPU phase and a styrene-butadiene-styrene (SBS) phase [106,107]. The introduction of the double-percolation structure, consisting of SBS phases incompatible with TPU, lowered the percolation threshold of the CNT/TPU@SBS strain sensor to as low as 0.38 wt.%, while the conductivity of 1%-CNT/TPU@SBS (4.12 × 10^−3^ S/m) was two orders of magnitude higher compared to that of 1%-CNT/TPU (3.17 × 10^−5^ S/m) [69]. This double-percolation structure also provides the 1%-CNT/TPU@SBS strain sensor with a wide strain detection range (0.2–100%), ultra-high sensitivity (maximum GF of 32,411 at 100% strain), fast response time (214 ms), and stability (500 loading/unloading cycles).

Since CNTs usually cluster together in the TPU matrix due to van der Waals forces, this affects the overall conductivity of the flexible strain sensors. Therefore, improving the dispersion state of CNTs in the TPU matrix is beneficial for preparing flexible strain sensors with higher electrical conductivity [108]. The most economical and common strategy to improve the dispersion of CNTs is the ultrasonic treatment method. This involves ultrasonically dispersing CNTs into a suspension by adding CNTs to deionized water and then immersing TPU nanofiber membranes into it for further ultrasonic treatment to prepare CNT/TPU composite nanofiber membranes [70]. CNT/TPU strain sensors based on the composite nanofiber membranes offer an excellent combination of high sensitivity (maximum GF of 1571), superior tensile strength and toughness (stresses greater than 24 MPa and strains greater than 400%), outstanding durability (10,000 cycles at 10% strain), and a wide operating tensile range (0–400%). Some researchers prepared a CNT/TPU composite nanofiber yarn with an elongation at a break of up to 476% by ultrasonically mixing TPU dissolved in a CNT dispersion and applying a multi-needle liquid bath electrospinning technique to the mixture [25]. Ultrasonic mixing followed by electrospinning significantly improved the dispersion of CNTs in the composite nanofiber yarn. A highly conductive dip-coated CNT/TPU composite nanofiber yarn (1.02 kΩ/cm) was developed by simply dip-coating the CNT ink. The yarn-based flexible strain sensor exhibits a high relative resistance change (440%) with satisfactory linearity and repeatability (1250 cycles) at 140% strain. Building on this, some researchers developed a TPU/CNT composite nanofibrous tube consisting of wavy and oriented fibers coated with carboxylated CNTs using spraying and ultrasound-assisted deposition [71] (Figure 4c). The combination of spraying and ultrasound-assisted deposition resulted in 12 wt% CNT deposition on the composite nanofiber tube, forming an excellent conductive network with electrical conductivity up to 0.01 S/cm. The as-prepared strain sensors exhibit a wide strain sensing range of up to 760%, high sensitivity (GF of 57), low detection limit (2%), and fast response time (45 ms) while maintaining reliable long-term stability and durability. Moreover, TPU/MWCNT strain sensors were prepared by filtering an aqueous dispersion of MWCNTs through electrospun TPU nanofiber membranes, drying them, welding them directly onto T-shirts, and encapsulating them with a thin silica gel layer [109]. Flexible strain sensors prepared with KMnO4-oxidized MWCNTs have a GF of up to 46 at an applied strain of 10%. In addition, there are other ways to achieve better conductivity in flexible strain sensors, such as increasing the level of dispersion by incorporating other conductive materials that aid in the dispersion of CNTs. This approach will be explained later when introducing other conductive materials for electrospun TPU strain sensors.

This section examines the application of one-dimensional carbonaceous material, specifically CNTs, in electrospun TPU flexible sensors. CNTs, including single-walled and multi-walled varieties, are praised for their aspect ratio and low percolation threshold, which contribute to the formation of stable conductive pathways in TPU substrates. The section details methods to improve CNT dispersion and loading, such as using polydopamine coatings and ultrasonic treatment, resulting in enhanced sensor sensitivity, response/recovery times, and durability.

#### 3.1.3. Two-Dimensional Carbonaceous Material—Gr

Graphene (Gr) is a two-dimensional carbonaceous material consisting of a single layer of sp^2^-hybridized carbon atoms with intrinsic conductivity typically between 10^3^ and 10^6^ S/cm [110]. The laminated Gr nanosheets have a high surface area, large specific surface area, and two-dimensional structure, making it easier to form a layer-connected planar-to-planar conductive network in the TPU substrate [111]. Simultaneously, the excellent electrical conductivity of Gr lowers the percolation threshold, reducing the impact on the tensile properties of TPU composite nanomaterials [112]. Moreover, the two-dimensional layered structure of Gr enables it to be tightly bonded to the TPU substrate, responding promptly to external stimuli and exhibiting excellent electrical sensitivity [113]. Therefore, Gr and its derivatives (e.g., reduced graphene oxide (rGO) and graphene oxide (GO)) are widely used as conductive fillers in electrospun TPU strain sensors due to their excellent electrical, thermal, mechanical, and sensing properties. 

A common strategy is to ultrasonicate the electrospun TPU nanofiber membrane in a mixed solution of rGO and sodium dodecyl sulfate (SDS), and then polymerize the treated composite nanofiber membrane in a DA solution to obtain a PDA/rGO/TPU flexible strain sensor [72]. Here, PDA containing hydrophilic groups (-OH and -NH_2_) improves the hydrophilicity and breathability of the composite nanofiber mats and prevents rGO from falling off the mats. PDA/rGO/TPU flexible strain sensors exhibit an excellent combination of properties, including high sensitivity (gauge factor of 6583 at strains above 140%), wide sensing range (0–150%), fast response time (100 ms), excellent durability (9000 cycles at 50% strain), and excellent gas permeability (water vapor evaporation rate of 38%). Building on this, some researchers explored 1H,1H,2H,2H-perfluorodecanethiol (PFDT) fluorination of PDA/rGO/TPU composite nanofibrous membranes by water vapor gasification [114]. PFDT fluoride treatment imparts superhydrophobicity so that the sensor, under any strain condition, repels water, acid, and alkali and exhibits corrosion resistance, with the contact angle increasing from 117° to 150°. This makes the sensor applicable to various harsh and complex working environments. 

In addition, some researchers have explored the construction of various novel structures to further enhance Gr’s conductivity in TPU flexible strain sensors. Researchers innovatively prepared thermoplastic polyurethane-based cellulose acetate composite nanomembranes (CA/TPU) with abundant mesopores by electrostatic spinning and then firmly anchored rGO and GO successively to them by ultrasonic impregnation to obtain an rGO/GO@CA/TPU flexible strain sensor [115] (Figure 5a). The porous structure, obtained by phase separation utilizing differences in solvent boiling point and solubility, has a high specific surface area and suitable pore size, greatly improving the adsorption degree and fastness of rGO and GO. The rGO on the surface of the CA/TPU composite nanofiber membrane interconnects to form a good conductive network, while GO prevents the overlapping of rGO during the stretching process, thereby improving the sensitivity and stability of the sensing electrical signal. This enables the rGO/GO@CA/TPU flexible strain sensor to achieve excellent sensitivity (gauge factor of 3.006) at extremely low strain (0.5%). However, increases in conductivity, sensitivity, and response time are often accompanied by decreases in ductility and sensing range. The simultaneous introduction of rGO and GO reduces the tensile resilience of the composite nanofiber membrane, and the rGO/GO@CA/TPU flexible strain sensor shows good testing results only in the low strain range (0.5–10%). Nevertheless, a team of researchers explored pressing and dragging layers of graphite nanoplates (GNPs) on a TPU substrate and then encapsulating them with polydimethylsiloxane (PDMS) as a protective layer to create a new crack-wrinkle structure (CWS) ultrasensitive strain sensor [73] (Figure 5b). The area proportions of cracked and wrinkled structures in this CWS flexible strain sensor are 31.8% and 9.5%, respectively. According to crack extension theory, when the CWS flexible strain sensor is subjected to a small stretch, the cracks in the GNP layer break, leading to a change in the contact area, which rapidly causes a significant change in resistance. This allows the CWS strain sensor to exhibit ultra-high sensitivity at low strains (gauge factor of 750 at 24% strain), ultra-low detection limits (a strain of 0.1%), and ultra-short response times (90 ms). Even though the introduction of the wrinkle structure reduces the irrecoverable damage of cracks and improves the recoverability and stability of the CWS flexible strain sensors to some extent, the CWS flexible strain sensors can still only work properly at a lower strain range (0.1–24%). To balance sensitivity and strain range, researchers prepared porous TPU–PEO composite nanofiber membranes by water-washed electrospinning and then stimulated them with anhydrous ethanol to obtain a porous curled network structure [74] (Figure 5c). The effective combination of porous and convoluted structures ensures high sensitivity (gauge factor of 4047.5 at 350% strain) of the composite nanofiber membrane after ultrasonic treatment with GNPs while achieving a wide strain operating range (0–350%). The TPU/GNP flexible strain sensor maintains excellent stability and durability after long-term operation (10,000 cycles).

Gr, a two-dimensional carbonaceous material, is explored in this section for its use in TPU strain sensors. Gr’s layered structure and conductivity enable the creation of a planar conductive network in TPU, offering high sensitivity and a low percolation threshold. The section describes strategies to integrate Gr with TPU, such as ultrasonication and polymerization, to produce sensors with high sensitivity, fast response, and durability. It also discusses the construction of novel structures to enhance Gr’s conductivity and the balance between sensitivity and strain range in sensor design.

#### 3.1.4. Multi-Dimensional Hybrid Carbonaceous Materials

The conductive carbonaceous materials of different dimensions dispersed in the TPU substrate have different aggregation degrees and construct conductive networks with different structures. The various structures of the conductive network further affect the conductivity, percolation threshold, and tensile properties of the TPU composite nanomaterials, thus reflecting the differences in the sensing properties of the electrospun TPU flexible strain sensors in terms of tensile properties, sensitivity, sensing range, and other properties [75,116]. Generally speaking, the conductive network constructed by carbonaceous conductive materials with lower dimensionality is fragile in structure, easily fractured when stretched, and has higher electrical sensitivity, but can only work in a narrower strain range [117]. On the other hand, carbonaceous conductive materials with high dimensionality have high aspect ratios, which are easy to overlap and entangle with each other in the TPU substrate, and the constructed conductive network is more stable and solid, showing a lower percolation threshold and a wider sensing range [76]. Hence, researchers have explored a variety of new ways to combine and tune conductive carbonaceous materials of different dimensions to combine their strengths, leading to the development of electrospun TPU flexible strain sensors with simultaneous high conductivity, high sensitivity, and wide sensing range.

One research group utilized ultrasonic treatment to adsorb both CB and CNTs onto electrospun TPU nanofiber membranes, subsequently modifying them with PDMS to achieve low surface energy and create a superhydrophobic PDMS/(CB+CNTs)/TPU flexible strain sensor [77] (Figure 6a). CB, as the main filler, aggregates into clusters, while CNTs, serving as auxiliary fillers, intersperse between CB clusters to bridge the gaps, forming a more stable hybrid conductive network with “point-line” contact [78,79]. Meanwhile, PDMS acts as a binder, bonding the CB, CNTs, and TPU together, significantly improving the overall ductility of the composite nanofiber interface. This enables the PDMS/(CB+CNTs)/TPU flexible strain sensors to exhibit excellent sensitivity (GF up to 49,863.5 in the 431.5–437.9% strain range) and a wide strain operating range (0–437.9%). Another research group prepared a TPU/5CNT/5Gr flexible strain sensor by simultaneously depositing carboxylated CNTs and Gr on electrospun TPU nanofiber membranes in a 5:5 ratio via vacuum-assisted deposition and ultrasonication [80] (Figure 6b). Due to the π-π interaction, Gr nanosheets tend to stack, affecting the tensile properties of TPU composite nanofiber membranes [118]. However, the synergistic effect between Gr and CNTs mitigates this issue. Gr acts as a spacer to separate the intertwined CNTs, while the long and zigzag CNTs inhibit the overlapping of Gr nanosheets and fill the gaps between them, forming a three-dimensional electrically conductive network with “line-plane” contacts [119]. This design allows the TPU/5CNT/5Gr flexible strain sensor to simultaneously exhibit a wide operating range of 172% and a high strain measurement factor of approximately 217.

In addition, some researchers developed a highly stretchable and conductive MCN/TPU composite nanofiber yarn (MCN/TPU-NY) by combining CB, CNTs, and Gr ternary carbon nanomaterials with electrospun TPU nanofiber yarns via multi-needle liquid bath electrospinning and twisting techniques [81] (Figure 6c). By comparing the effects of different MCN compositions on sensor performance, it was found that CB-based fillers could achieve high electrical conductivity and maintain satisfactory mechanical properties under high loading (24 wt.%) conditions. When the concentration reaches the networking threshold (3 wt.%), 1D CNTs as auxiliary fillers can improve mechanical properties and strain operating range but weaken electrical conductivity. On the other hand, 2D Gr as auxiliary fillers can enhance electrical conductivity and sensitivity but reduce mechanical properties. The 24CB/3CNT/3GR flexible strain sensor, prepared by combining these three carbon nanofillers to construct a ternary conductive network, exhibits excellent sensing performance with high sensitivity (GF of 17.74 at 100% strain), a wide detection strain range (0–227%), fast response time (220 ms), and strong durability (5000 cycles at 50% and 100% strain).

This subsection explores the synergistic effects of combining carbonaceous materials of different dimensions in TPU strain sensors. Hybrid networks of zero-dimensional CB, one-dimensional CNTs, and two-dimensional Gr are designed to enhance conductivity, sensitivity, and strain range. Techniques like ultrasonic treatment and vacuum-assisted deposition are employed to achieve stable and efficient conductive networks. These hybrid structures address the limitations of individual fillers, offering improved sensor performance across various metrics.

### 3.2. MXene

Since its first synthesis in 2011, MXene (two-dimensional transition metal carbides and carbonitrides) has been widely used in applications such as wearable devices, energy storage, catalysis, and electromagnetic interference shielding [120]. Although more than 30 MXene materials have been synthesized in laboratory settings, the main form of MXene used in electrospun TPU flexible strain sensors is Ti_3_C_2_T_x_. Here, T_x_ denotes various surface terminal groups (e.g., O, OH, F, and Cl) obtained by different synthesis methods [121]. By tuning the functional groups on the surface of MXene (Ti_3_C_2_T_x_), it can be effectively combined with various conductive materials on the TPU substrate to form a synergistic effect, leading to the preparation of composite nanofibers with higher conductivity [122]. Furthermore, MXene (Ti_3_C_2_T_x_) itself has excellent electrical conductivity, hydrophilicity, and mechanical properties, which are favorable for its dispersion in the electrospun TPU matrix, thus enabling the preparation of electrospun TPU flexible strain sensors with both excellent conductivity and ductility [123].

A basic strategy is to deposit the MXene sensing layer on the electrospun TPU nanofiber membrane by electrostatic spraying to prepare a sandwich-structured resistive MXene/TPU flexible strain sensor [124]. During the electrostatic spray deposition process, the agglomeration of MXene is reduced due to the Coulomb repulsion between conductive particles with the same charge. Meanwhile, the electrostatic spray deposition results in better adhesion of the prepared MXene sensing layer to the electrospun TPU nanofiber membrane compared with other deposition processes, leading to more uniform stress distribution during stretching and a wider strain operating range (extended to 70%) of the prepared MXene/TPU flexible strain sensor. On this basis, some researchers synthesized dopamine to form a polydopamine coating on electrospun TPU nanofiber membranes, then ultrasonically loaded MXene to prepare a high-performance MXene/PDA/TPU flexible strain sensor [82]. The PDA coating not only improves the adhesion between the MXene and the TPU but also combines with the MXene to enhance the EMI shielding performance of the prepared MXene/PDA/TPU flexible strain sensors (absolute effective shielding value of 8383.33 dB·cm^2^·g−^1^). The MXene/PDA/TPU flexible strain sensor also exhibits a low detection threshold (0.1%), high sensitivity (GF up to 2600 at 200% strain), a wide strain operating range (as wide as 200%), and fast response/recovery times (77/186 ms). Furthermore, other researchers prepared a TPU/MXene/TPU-BNNS (TMTB) flexible sandwich strain sensor with advanced thermal regulation by introducing a TPU/boron nitride nanosheet (BNNS) thermally conductive layer assembled with the electrospun TPU substrate and the MXene sensing layer [83] (Figure 7a). The high orientation and strong van der Waals interactions between BNNS and TPU in the TPU/BNNS thermal conductivity layer facilitate the construction of effective thermal conductivity channels and reduce phonon scattering between the interacting units, greatly improving the thermal conductivity at a low filler permeability threshold (thermal conductivity up to 1.5 W·m−^1^·K−^1^). This unique structure simultaneously endows the prepared sensor with a wide strain operating range (0–100%), ultra-high GF (2080.9), and reliable biocompatibility and antimicrobial properties.

For MXene/TPU composite nanofiber membranes, because of the huge Young’s modulus difference between the MXene and TPU matrices, the MXene conductive layer can be easily crushed into large fragments after the pre-stretching treatment to build an ultrasensitive microcracked structure [125]. However, the collapse of the microcracks under high strain leads to severe distortion and disintegration of the conductive coating. This causes the flexible strain sensors based on MXene/TPU composite nanofiber membranes to exhibit relatively high GF values and narrow strain sensing ranges. To address this issue, a common strategy is to utilize the synergistic effect of MXene and CNTs to construct excellent conductive networks with both high conductivity and strong tensile resistance in the TPU matrix [126]. The introduction of CNTs changes the direction of tensile stress transfer in MXene conductive coatings, resulting in smaller MXene fragments and denser microcrack structures. Meanwhile, CNTs can effectively bridge the MXene microcrack fragments, weakening the microcrack effect under tension and lengthening the separation process of neighboring microcrack fragments [127]. A research group has developed a MXene/CNT/TPU flexible strain sensor with a typical bilayer-conductive structure by self-assembling MXene and CNTs onto porous electrospun TPU nanofiber membranes via a vacuum filtration deposition process [84] (Figure 7b). During stretching, the brittle, densely stacked MXene upper lamellae crack, while the flexible MXene/CNTs-decorated TPU lower fibers connect the crack gaps, ensuring that the synergistic conductive network remains intact. This enables the MXene/CNT/TPU flexible strain sensors to exhibit high sensitivity and a wide operating range (GF values of 37.5, 289.5, 1070.1, and 2911 in the 0–50%, 50–150%, 150–250%, and 250–330% strain ranges, respectively), as well as a fast response time (80 ms) and superior durability (over 2600 cycles at 50% strain). On this basis, another research group developed a sandwich-structured MXene-CNT/TPU/Ecoflex flexible strain sensor by simultaneously coupling CNTs and MXene on wave-structured electrospun TPU nanofiber membranes through vacuum suction filtration and Ecoflex encapsulation [85] (Figure 7c). The low viscoelasticity of Ecoflex, combined with the wave-structured aligned TPU nanofiber network, was utilized to reduce the hysteresis (as low as 50 ms) of the prepared sensor when stretched in the vertical fiber direction. The MXene-CNT/TPU/Ecoflex flexible strain sensor also exhibits a low detection limit (0.1% strain), high sensitivity (GF of 1719.2 at 251% strain), and a wide sensing range (0.1–251% strain). Inspired by the pearl multilayer interaction structure, some researchers have developed a pearl-simulated multilayer structure of MGO/TPU flexible strain sensors by adding MXene/rGO (MGO) solution to multilayer TPU electrospun mats [128]. The Ti-O-C covalent bond formed by dehydration and nucleophilic substitution reactions between MXene and rGO strengthens the connection between MXene and rGO, enhancing the stretchability and sensing stability of the MGO/TPU flexible strain sensor (more than 5000 tensile cycles). Meanwhile, the pearl-like simulated multilayer structure allows the tuning of the operating range of the MGO/TPU flexible strain sensor (from 80% to 220%) by varying the mass ratio of MXene to rGO and the number of layers of TPU electrospun felts. Thanks to the synergy between MXene and rGO and the special design of the pearl-simulated multilayer structure, the MGO/TPU flexible strain sensor also demonstrates high sensitivity (GF > 84,326), fast response time (70 ms), and a low detection limit (0.05% strain).

MXene, a two-dimensional transition metal carbide, is discussed in this section for its role in electrospun TPU flexible strain sensors. MXene’s conductivity, hydrophilicity, and mechanical properties make it suitable for creating composite nanofibers with enhanced conductivity and ductility. The section outlines strategies for depositing MXene layers and enhancing their dispersion in TPU, leading to sensors with high sensitivity, a wide strain range, and improved thermal regulation.

### 3.3. Metallic Materials

#### 3.3.1. Silver Nanomaterials

Conventional metallic materials are not suitable for direct use in flexible strain sensors due to their hard texture and poor ductility; however, metallic nanomaterials are explored as conductive fillers due to their nanoscale size and are immobilized on electrospun TPU substrates through various processes (intercalation, coating, etc.) to create flexible strain sensors with excellent performance [129,130]. Among many metal nanomaterials, silver nanomaterials, such as silver nanowires (AgNWs) and silver nanoparticles (AgNPs), have been widely explored for the creation of high-performance flexible strain sensors by constructing various structures or synergizing with other conductive fillers (CNTs, MXene, etc.) in electrospun TPU substrates due to their high electrical conductivity (6.3 × 10^7^ S/cm), excellent optical transparency, and appropriate yield strength [131,132,133].

Compared with one- or two-dimensional conductive materials with high aspect ratios, the nanoparticle morphology of AgNPs offers greater flexibility, which makes it more conducive to achieving reversible nanoscale recovery [134]. However, the high density of AgNPs and the poor adhesion between them and the electrospun TPU substrate make them susceptible to detachment by external forces, which affects the tensile and sensing stability of the prepared flexible strain sensors [135]. To solve this problem, some researchers developed a TPU/PDA/AgNP/SR flexible strain sensor by electroless deposition (ELD) of AgNPs on the surface of wavy electrospun TPU mats obtained by ethanol treatment with PDA and silicone rubber (SR) assistance [136]. The ethanol-treated electrospun TPU mats spontaneously curled to form a wavy structure, which, combined with the modification and assistance of PDA and silicone rubber (SR), significantly enhanced the deposition efficiency and fastness of AgNPs and improved the tensile and sensing stability of the sensors. Considering the high cost of DA, some other researchers suggested depositing AgNPs on the surface of electrospun TPU nanofiber felt with the aid of a mixed coating of tannic acid (TA) and hydrolyzable 3-aminopropyltriethoxysilane (APTES) (TA-APTES coating) to construct a composite conductive nanofiber material with a hierarchical AgNP shell and TPU nanofiber core microstructure [137]. The modification of the TA-APTES hybrid coating not only promotes the adsorption and immobilization of AgNPs on the surface of the electrospun TPU nanofiber mats but also enhances the interactions between the AgNPs and the electrospun TPU substrate, resulting in TPU@(TA-APTES)@AgNP flexible strain sensors that exhibit both high tensile properties (maximum strain of 565%) and high sensitivity (GF up to 6886). Moreover, some researchers have explored the use of the synergistic effect of zero-dimensional AgNPs and one-dimensional CNTs, two-dimensional MXene, to construct hybrid multidimensional conductive networks on electrospun TPU substrates, aiming to develop flexible strain sensors with excellent conductivity and stretchability [138,139]. A research group prepared a bilayer conductive TPU/Ag/CNT flexible strain sensor comprising a conductive layer of AgNPs and a conductive layer of CNTs by simultaneously anchoring AgNPs and CNTs on electrospun TPU nanofiber membranes via magnetron sputtering and ultrasonic treatment [86] (Figure 8a). During the stretching process, the rigid AgNP conductive layer continues to produce cracks until it deforms to form an “island structure”; and the high aspect ratio CNTs act as a “bridge” to connect the cracks in the “island structure” of the AgNP conductive layer, synergistically forming a more stable and stretch-resistant conductive network, thereby expanding the response range of the prepared strain sensor (up to 604% strain). The synergistic effect of AgNPs and CNTs also endows the TPU/Ag/CNT flexible strain sensors with excellent sensing performance, including high sensitivity (strain coefficient up to 6834), fast response and recovery time (122 ms/164 ms), and extremely low detection limit (0.1%). Notably, the oxidation of silver nanomaterials in air affects the stability and durability of flexible strain sensors during their application [140]. Therefore, a research group designed a dual conductive layer of AgNP/CNT/PDA/TPU flexible strain sensor leveraging the synergistic effect of AgNPs and CNTs by anchoring PDA on the surface of electrospun TPU fiber mats and then decorating CNTs and AgNPs by ultrasonication and in situ polymerization, respectively [141] (Figure 8b). The introduction of PDA not only improves the interaction between CNTs, AgNPs, and the interface with the electrospun TPU substrate, thereby broadening the detection range of the prepared sensors (up to 640%); it also effectively protects the AgNPs from oxidation during the working process, ensuring the response stability and long-term durability of the prepared sensors. Another research group modified the electrospun TPU nanofiber membrane by ultrasonic coating with MXene (Ti_3_C_2_T_x_) followed by loading AgNPs by chemical deposition to prepare an AgNP/MXene/TPU flexible strain sensor utilizing the synergistic effect of AgNPs and MXene [142]. MXene (Ti_3_C_2_T_x_) not only enhances the bonding stability and homogeneity between AgNPs and the electrospun TPU substrate through the rich chemical bonding on its surface but also disperses in the conductive network of AgNPs, acting as a “bridge” to enhance the electrical conductivity of the conductive network of AgNPs and its resistance to stretching. This enables AgNP/MXene/TPU flexible strain sensors to exhibit excellent EMI shielding performance (up to 87.31 dB at 8–12 GHz) as well as antimicrobial and hydrophobic properties, indicating great potential for applications in complex environments.

AgNWs have excellent thermal conductivity and chemical stability, but their unsatisfactory stretchability and poor adhesion to flexible substrates are important limitations to be addressed when applying them to electrospun TPU flexible strain sensors [143]. To address these limitations, and inspired by the convoluted structure of climbing plants, some researchers prepared a layered interlaced helical conductive yarn with both high mechanical elasticity and electrical conductivity by entangling the AgNW/MXene dimensional synergistic conductive network hierarchically with a network of electrospun TPU nanofibers [87] (Figure 8c). The high-aspect-ratio 1D AgNWs bridge the 2D MXene into a multidimensional synergistically interconnected conductive network, which intersperses with the electrospun TPU nanofiber network to form a unique hierarchical interlocking effect. This provides a robust channel for efficient electron/phonon transport in the layered helical conductive yarns, enabling the conductive yarns to exhibit high electrical conductivity (1.12 × 10^5^ S/m) even under large mechanical strain (300%). The AgNW/MXene/TPU flexible strain sensors developed based on this layered helical conductive yarn have excellent electromechanical properties with a stable GF (1.7) even in the 600–1000% strain range.

This section focuses on silver nanomaterials, such as silver nanowires and nanoparticles, as conductive fillers in flexible strain sensors. The high electrical conductivity and optical transparency of silver nanomaterials make them ideal for sensor construction. The section discusses methods to improve the adhesion and dispersion of silver nanomaterials in TPU substrates, as well as the use of synergistic effects with other conductive materials to create multidimensional conductive networks for enhanced sensor performance.

#### 3.3.2. Liquid Metal

Liquid metals (LM), such as eutectic gallium-indium and gallium-indium-tin alloys (GaIn and GaInSn), are cohesive, electrically conductive liquids with melting points close to room temperature [144]. Liquid metal not only overcomes the limitations of traditional metals, such as high Young’s modulus and poor ductility, but also avoids the problems of high toxicity, low conductivity, and low vapor pressure associated with ionic liquid solvents [145]. It exhibits excellent properties such as low Young’s modulus, fluidity, low viscosity, high conductivity, and low toxicity, and has significant potential for applications in electrospun TPU flexible strain sensors [146].

A commonly used approach is to create deformable conductive fibers (DCFs) by injecting LM into hollow electrospun TPU nanofibers [147]. Due to the thermoplasticity of TPU and the high electrical conductivity of LM, DCFs can be molded into complex 2D or 3D shapes by heating at low voltage to the two thermoplastic transition points of TPU. Additionally, leveraging LM’s solid-liquid phase transition at its melting point, DCFs demonstrate variable shape memory properties at these transition points. The manufactured spiral DCF flexible strain sensors show excellent electrical performance (resistance change <2%) at a maximum strain of 3300%. However, LM potentially detaches from the hollow of the electrospun TPU nanofibers under external force, thus affecting the electrical conductivity of the composite fibers. Therefore, some researchers explored constructing a flexible 3D conductive network by coating LM droplets onto electrospun TPU nanofiber membranes via mechanical stretching to prepare an LM–TPU flexible strain sensor [88]. LM droplets are suspended between electrospun TPU nanofibers and coalesce into a conductive network. This enables the LM–TPU flexible strain sensors to exhibit an array of desirable properties, including a wide strain range (0–200%), good biocompatibility (can be used directly on the skin), and excellent stability and durability (9000 cycles). However, the simple coating renders the sensor less sensitive (GF of only 0.2 at 200% strain), and the lack of protection makes it susceptible to damage by the external environment, limiting its mass production and application. To solve these problems, some researchers proposed preparing ultra-high stretch, low-cost TPU/LM/NFs nanoyarns by printing LM onto the surface of electrospun TPU nanoyarns followed by encapsulation with nanofibers to form a protective layer [89]. The patterned printing of LM onto the surface of electrospun TPU nanoyarns makes it smooth, enhances interfacial interactions between the LM and TPU substrate, and improves the conductivity of the prepared TPU/LM/NFs flexible strain sensor (GF increased to 6 at 200% strain), while the protection provided by the outer nanofibers allows the sensor to exhibit high mechanical strength (40 MPa) and a wide sensing range (548%).

Liquid metals, with their low melting points and high conductivity, are explored in this section for their potential in electrospun TPU flexible strain sensors. The section describes approaches to creating deformable conductive fibers and flexible strain sensors using liquid metals, highlighting their biocompatibility, stability, and durability. It also discusses the challenges of maintaining electrical conductivity and environmental protection in liquid metal-based sensors.

### 3.4. Conductive Polymers

Conductive polymers belong to a class of polymers with exceptional electrical properties, showing tunable conductivity ranging from semiconductors to metallic conductors due to their intrinsic delocalized π-electrons [148]. Conductive polymers are also being explored to improve the sensitivity, sensing range, and durability of electrospun TPU flexible strain sensors because of their low cost, ease of synthesis, long-term stability, and non-toxicity [149]. Currently, typical conductive polymers like polypyrrole (PPy), poly(3,4-ethylenedioxythiophene):poly(styrenesulfonic acid) (PEDOT:PSS)), and polyaniline (PANI) are extensively used for the preparation of high-performance electrospun TPU flexible strain sensors.

PPy possesses excellent electrical conductivity, thermal stability, ease of preparation, and biocompatibility, making it ideal for direct use as a conductive filler in electrospun TPU flexible strain sensors [150]. Some researchers proposed preparing a fluorescent visible yarn strain sensor (SCFY strain sensor) based on a cured bead and crack structure by loading polydopamine (PDA) and polypyrrole (PPy) to create in situ designed cracks after heterostructurally designing a PDMS cured bead on the surface of the fluorescent fiber of TPU-TPE [90] (Figure 9a). The functional groups (e.g., ketone group and hydroxyl group) in PDA can form hydrogen bonds with N atoms in PPy as hydrogen bond donors or acceptors, which enhances the fastness of the conductive layer, thereby widening the working strain range (0–184%) of the prepared sensors. The PDMS solidified liquid beads and fibers exhibited stress mutation at the interface, causing more cracks in the PPy conductive layer under stress, thereby effectively enhancing the sensitivity of the sensor (GF value of 58.9 in the strain range of 143–184%). Moreover, the presence of TPE caused the cracks to emit blue fluorescence under UV lamp irradiation, enabling fluorescence visualization of the cracks.

Poly(3,4-ethylenedioxythiophene) (PEDOT), prepared by substituting thiophene rings at positions 3 and 4, demonstrates high corrosion resistance and stability in complex, harsh environments [151]. PEDOT also exhibits cost-effectiveness, superior electrical conductivity, and higher transparency than other conductive polymers [152]. However, this conductive polymer is insoluble in water or organic solvents, making it difficult to apply to electrospun TPU flexible strain sensors via impregnation or spin-coating techniques. Therefore, a negatively charged water-soluble component (PSS) was used to stabilize the positively charged PEDOT dispersion, resulting in the preparation of poly(3,4-ethylenedioxythiophene):poly(styrenesulfonic acid) (PEDOT:PSS)), which is easily applied to electrospun TPU flexible strain sensors [153]. A research team prepared a composite nanofiber membrane with a three-dimensional conductive network skeleton by dip-coating a mixed conductive filler of carboxylated multi-walled carbon nanotubes (C-MWCNT) and poly(3,4-ethylenedioxythiophene):poly(styrenesulfonic acid) (PEDOT:PSS)) with synergistic conductive effects onto an electrospun TPU nanofiber membrane substrate [91] (Figure 9b). Due to the Young’s modulus difference between the hybrid conductive layer and the electrospun TPU substrate, a microcrack structure was constructed on the surface of the composite nanofiber membrane by pre-stretching, and the strain sensitivity of the glycol-treated e-CPT9/1 flexible strain sensor (mass ratio of C-MWCNT to PEDOT:PSS is 9/1) was efficiently tuned by varying the density of the microcracks (GF = 6008.3). Meanwhile, the detection range (0.5–680%) of the e-CPT9/1 flexible strain sensor was broadened due to the synergistic effect of C-MWCNT bridging adjacent PEDOT crack fragments. In addition, the prepared e-CPT9/1 flexible strain sensor also exhibits remarkable “photo-thermal-electrical” conversion capability due to the excellent photo-thermal and thermoelectric effects of C-MWCNT and PEDOT.

Polyaniline (PANI) is currently the most studied conductive polymer due to its strong polarity, low cost, high processability, environmental stability, and unique doping/de-doping mechanism [3]. The strong polarity of PANI induces high electrical conductivity, but it also leads to unsatisfactory elasticity of PANI [154]. Combining PANI with TPU improves the elasticity, but the strong polarity limits the ease of combination. In situ polymerization is a good method to combine electrospun TPU nanofiber membrane and PANI explored by many researchers [155]. A research group explored the simultaneous in situ polymerization of MWCNT and PANI on electrospun TPU nanofiber membranes to prepare a PANI-TPU/MWCNT flexible strain sensor with excellent performance [33]. Due to the loading of PANI and the good dispersion of MWCNT in the TPU matrix, PANI and MWCNT synergistically constructed a multidimensional hybrid conductive network with both high conductivity and strong stretchability. In addition, another research group assembled a flexible wearable rGO/PANI/TPU strain sensor (GPTSS) prepared by decorating polyaniline (PANI) nanoparticles bridged by reduced graphene oxide (rGO) nanosheets onto electrospun TPU nanofiber mats via in situ polymerization and ultrasonic treatment [92] (Figure 9c). The in situ polymerization of PANI nanoparticles coated on the electrospun TPU pads improved the interfacial adhesion to some extent and enhanced the deposition efficiency of rGO. The ultrasonic treatment introduces rGO nanosheets to bridge the PANI nanoparticles so that the synergistically constructed hybrid conductive network is easy to recover in time after damage. This enabled the preparation of a flexible GPTSS that simultaneously exhibits a wide strain range (0.1–300% strain), high sensitivity (GF up to 3000.2), fast response time (90 ms), and excellent long-term durability. In addition, the GPTSS is also able to accurately monitor NH3 (5 ppm) due to the H+-doped structure of the PANI nanoparticles.

Conductive polymers, such as polypyrrole, PEDOT:PSS, and polyaniline, are examined in this section for their use in enhancing the performance of electrospun TPU flexible strain sensors. These polymers offer tunable conductivity, low cost, and stability. The section details methods of incorporating conductive polymers into TPU substrates, such as in situ polymerization and the creation of synergistic conductive networks, resulting in sensors with improved sensitivity, response times, and durability.

## 4. The Application of Electrospun TPU Strain Sensors

Overall, flexible strain sensors based on electrospun TPU have potential applications in motion detection, health monitoring, and human–computer interaction due to their excellent sensitivity, wide strain operating range, and fast response time with excellent stability and durability [156,157,158].

### 4.1. Motion Detection

In the era of IoT, Industry 4.0, and big data analytics, there is an increasing demand for real-time monitoring of human activities, which has spurred the development and application of more flexible and sophisticated wearable strain sensors in the domain of motion monitoring [71,97,128]. These sensors are designed to cater to the diverse spectrum of human movement, ranging from large-scale joint movements such as those of the hands, arms, and legs, to the minute details of facial expressions and physiological activities like smiling, blinking, and swallowing [159]. A research team, leveraging the synergies of IoT and big data, has engineered a CNT/F-TPU flexible strain sensor that boasts exceptional stretchability and heightened sensitivity [68]. This sensor, as depicted in Figure 10a, when affixed to the wrist, elbow, and knee of a subject, captures continuous, periodic, and stable sensing signals even as these joints undergo bending. This capability underscores the sensor’s proficiency in monitoring a broad spectrum of human body movements. Furthermore, the CNT/F-TPU strain sensor, when attached to the subject’s finger, demonstrates its ability to monitor even subtle movements by producing a regular response curve as the bending angle incrementally increases from 0° to 30°. Additionally, when positioned on the face and neck, the sensor mirrors the frequency of the tester’s facial expressions and swallowing actions, with a rapid increase in the ΔR/R_0_ value, thereby showcasing its potential to capture minute physiological changes. Meanwhile, another research collective has fabricated a highly stretchable and conductive CNT/TPU composite nanofiber yarn, ingeniously integrating it into an elastic self-adhesive bandage [25]. This smart sports bandage, as illustrated in Figure 10b, not only provides real-time guidance to beginners in sports like badminton by analyzing the peak characteristics of the relative resistance change (RRC) but also aids basketball players in refining their shooting postures and improving their scoring rates by sensing the elbow’s bending during the shooting motion. Moreover, when wrapped around the knee, the bandage’s sensor can monitor the strain and calculate the frequency and amplitude of leg movements, offering insights into the athlete’s performance and travel status. When applied to the chest and abdomen, the sensor functions as a real-time health monitor, tracking heart rate, respiratory rate, and exercise intensity, thereby ensuring safety during physical activities and preventing potential risks associated with arrhythmia or respiratory distress. These innovations, powered by IoT connectivity, are a testament to the seamless integration of advanced materials, sensor technology, and data analytics, paving the way for a new horizon in personalized health monitoring and sports performance enhancement in the context of Industry 4.0 [113,125,144]. 

### 4.2. Health Monitoring

In the burgeoning field of digital health, the advancement of electrospun TPU flexible strain sensors plays a pivotal role in health monitoring [91,96,131]. These sensors are engineered to exhibit heightened sensitivity to the nuanced physiological signals that are often encountered in diverse and challenging environmental conditions [161]. Given the prolonged skin contact of such wearable devices, their design must also prioritize breathability and biocompatibility to ensure comfort and safety during extended use [162]. Several researchers have delved into the innovation of a 3D woven structure of MXene/AgNW/TPU core yarn sensors, known as KMYS, which have been seamlessly integrated into an array of apparel through the application of inlay yarn guide technology [129]. As shown in Figure 10c, the KMYS has been skillfully incorporated into various flexible wearable devices such as pendants, elbow pads, wristbands, finger cuffs, socks, and face masks. These devices are capable of sensing physiological rhythms, including the pulses of the carotid, brachial, radial, digital, and ankle arteries, as well as respiratory movements. The KMYS demonstrates its versatility by not only detecting the pronounced pulse signals of the carotid, brachial, and radial arteries but also by accurately capturing the subtler pulse waves of the finger and ankle arteries. A 31-day pulse monitoring experiment conducted on a 30-year-old female subject revealed that the KMYS could reliably capture daily pulse waveforms over an extended period, providing a precise indication of cardiovascular health. Furthermore, the KMYS, when worn on the abdomen, can monitor breathing and pulse signals during sleep, offering a proactive approach to preventing life-threatening conditions such as sleep apnea. The sensor’s resilience is also evident in its ability to maintain stable pulse signal monitoring even when submerged in water, as demonstrated by tests where the KMYS was worn on the wrist and immersed. This capability underscores the sensor’s potential for capturing high-fidelity physiological signals for health detection across a multitude of complex application scenarios, thereby reinforcing its role in the realm of digital health monitoring [103,114,147].

### 4.3. Human–Computer Interaction

In the burgeoning landscape of AI and robotics, the quest for more sensitive and intelligent human–computer interaction is driving innovation. Electrospun TPU flexible strain sensors, known for their high conductive sensitivity and exceptional tensile recovery performance, are being explored as cutting-edge wearable devices that revolutionize the domain of human–computer interaction [163]. As depicted in Figure 10d, a TPU/CB flexible strain sensor has been developed for voice interaction [64]. This sensor, when mounted on the neck, adeptly captures the minute vibrations of the user’s vocal cords during speech. By analyzing the resultant response signal, it is possible to calculate the number of words spoken and the frequency of speech, thereby aiding the computer in speech recognition and enhancing the efficiency of human–computer interaction. Figure 10e introduces a TPU/IL ionogel fiber-like sensor designed for information encryption and transmission [160]. This sensor is integrated into a glove, transforming the human hand into a conduit for encrypted communication. By defining short and long finger bends as dots and dashes, respectively, Morse code is employed to encrypt letters or numbers [164]. These encrypted messages are then transmitted wirelessly to a cell phone, thereby broadening the horizons of advanced human–computer interaction. The integration of AI and robotics in these wearable sensors exemplifies the seamless fusion of technology and human capability, offering a new paradigm for secure, efficient, and intuitive communication between humans and machines. This innovation not only propels the field of human–computer interaction forward but also underscores the potential of AI and robotics in creating more responsive, intelligent, and interactive devices that can adapt to the user’s needs.

## 5. Conclusions and Outlook

In recent years, flexible strain sensors have been increasingly developed for applications in motion monitoring, health detection, human–computer interaction, and other areas of high interest. The TPU nanofiber material fabricated by electrospinning combines the advantages of both nanomaterials and fiber materials, with excellent characteristics such as extremely fine size, large specific surface area, high porosity, controllable structure, and functional modification by doping, making it very suitable for the preparation of flexible strain sensors as a substrate material in combination with various conductive fillers. Therefore, it is necessary to review the flexible strain sensors based on this special substrate material. This review paper introduces and discusses recent advances in electrospun TPU flexible strain sensors, focusing on summarizing strategies for introducing a variety of conductive fillers, including carbonaceous materials, MXene, metallic materials, and conductive polymers.

(1)As we stand on the precipice of a transformative era in the field of flexible strain sensors, the future holds a wealth of opportunities and challenges. Enhancing material properties is central to ongoing research, with scientists striving to achieve a more uniform distribution of conductive fillers within the microstructure of materials. Exploring novel composites and nanomaterials, such as carbon nanotubes, graphene, and metallic nanoparticles, is expected to significantly improve the sensitivity, response time, and durability of strain sensors.(2)The integration of electrospun TPU flexible strain sensors with AI and IoT presents a particularly promising avenue. AI algorithms can enable real-time data processing and analysis, leading to more precise health monitoring and predictive maintenance in industrial settings. IoT integration facilitates seamless data transmission, which is essential for remote monitoring and control systems in various applications.(3)Innovations in manufacturing techniques, such as the fusion of 3D printing with electrospinning, are set to revolutionize the development of electrospun TPU flexible strain sensors. This hybrid approach allows for the creation of sensors with complex geometries and tailored mechanical properties to meet specific application requirements.(4)Despite the optimistic outlook, challenges remain. The long-term stability and durability of sensors under real-world conditions are significant concerns. Sensors must be able to withstand environmental factors such as temperature fluctuations, humidity, and mechanical stress without performance degradation. Addressing these issues requires rigorous material testing and the development of robust encapsulation techniques.(5)Biocompatibility and safety are paramount, especially in applications where sensors come into direct contact with the human body. Extensive research is needed to ensure that sensor materials do not induce adverse reactions and are safe for long-term use. This includes understanding the interaction between sensor materials and biological systems at the molecular level.(6)Establishing standardized testing protocols and regulatory frameworks is critical as the technology evolves. Universally accepted standards are necessary to evaluate the performance, reliability, and safety of electrospun TPU flexible strain sensors, facilitating their commercialization and building consumer trust.(7)Cost remains a significant barrier to widespread adoption. Research into scalable manufacturing processes and the use of cost-efficient materials is essential for making these sensors more accessible to a broader market.(8)Overcoming these challenges necessitates a collaborative approach, involving experts from diverse fields such as materials science, electrical engineering, biomedical engineering, and data science. Interdisciplinary research will foster innovation and accelerate the development of electrospun TPU flexible strain sensors capable of meeting future demands.

In conclusion, while the future of electrospun TPU flexible strain sensors is promising, it requires a concerted effort to navigate the complexities of material development, technological integration, and practical implementation. By addressing the challenges head-on and fostering a collaborative research environment, we can unlock the full potential of these sensors and usher in a new era of smart and responsive technologies that enhance our lives and industries.

## Figures and Tables

**Figure 1 sensors-24-04793-f001:**
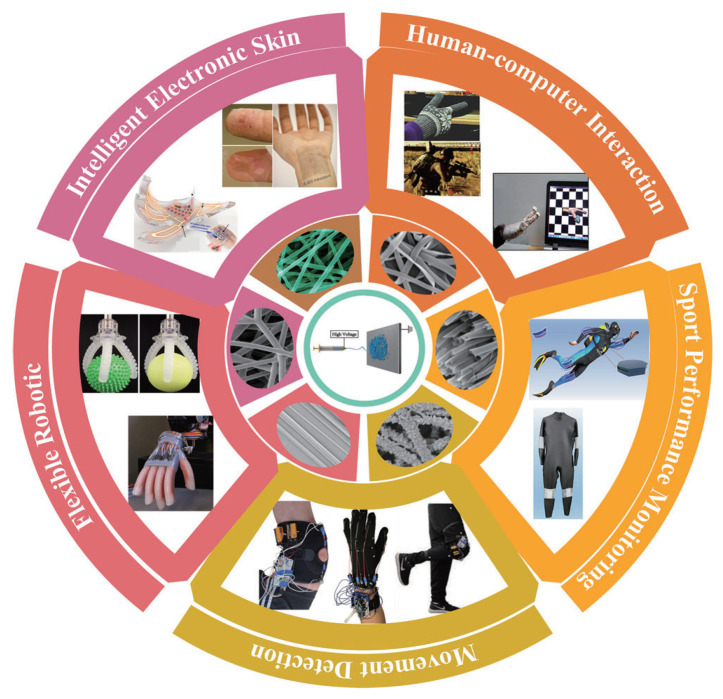
Electrospun flexible strain sensors and potential applications in intelligent electronic skin, human–computer interaction, sports performance monitoring, movement detection, and flexible robotic. Reprinted with permission from Ref. [9]. Copyright 2023, Wiley.

**Figure 2 sensors-24-04793-f002:**
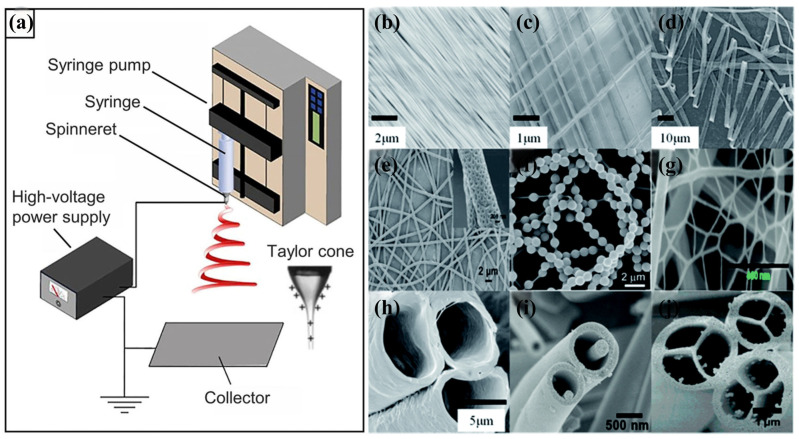
(**a**) Basic setup for electrospinning. Reprinted with permission from Ref. [42]. Copyright 2019, American Chemical Society. Different morphologies of electrospun polymer fibers: (**b**) uniaxially aligned, (**c**) biaxially oriented, (**d**) ribbon, (**e**) porous fibers, (**f**) Necklace-like, (**g**) nanowebs, (**h**) hollow, (**i**) nanowire-in-microtube, and (**j**) multichannel tubular. Reprinted with permission from Ref. [51]. Copyright 2014, the Royal Society Chemistry.

**Figure 3 sensors-24-04793-f003:**
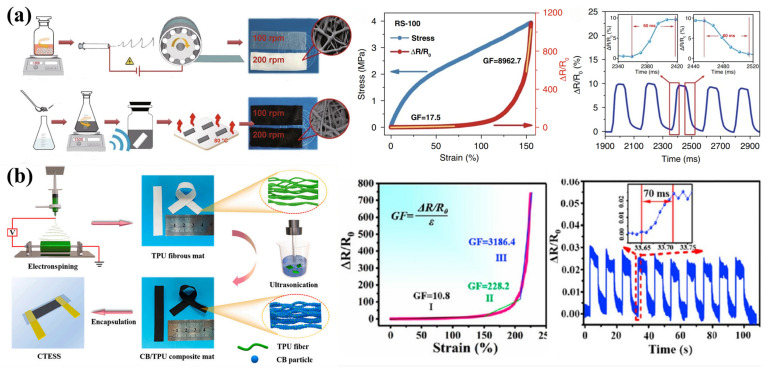
(**a**) Schematic illustration of the preparation process of CB/TPU strain sensor. ΔR/R_0_ of CB/TPU strain sensor toward the applied strain; the response time of CB/TPU strain sensor under 1% strain. Reprinted with permission from Ref. [64]. Copyright 2021, Springer. (**b**) Schematic illustration of the preparation process of CB/TPU/Ecoflex strain sensor (CTESS). ΔR/R_0_ of CTESS toward the applied strain; the response time of CTESS under 1% strain. Reprinted with permission from Ref. [65]. Copyright 2020, Elsevier.

**Figure 4 sensors-24-04793-f004:**
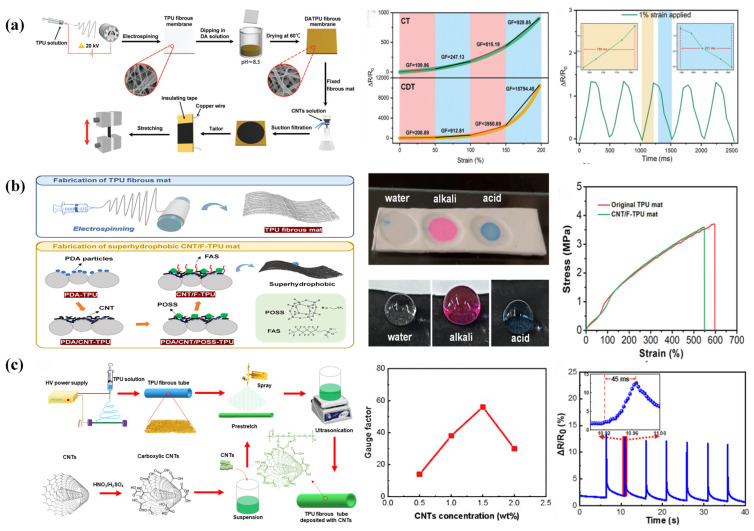
(**a**) Schematic illustration of the preparation process of a CNT/DA/TPU strain sensor; ΔR/R_0_ of CNT/TPU and CNT/DA/TPU strain sensor toward the applied strain; the response/recovery time of CNT/DA/TPU strain sensor under 1% strain. Reprinted with permission from Ref. [67]. Copyright 2023, Wiley. (**b**) Schematic illustration of the preparation process of CNT/F-TPU strain sensor; photographs of deionized water, 1 M alkaline, and 1 M acidic droplets on a raw TPU surface (top) and the CNT/F-TPU surface (bottom); comparison of the stress−strain curves of TPU and CNT/F-TPU. Reprinted with permission from Ref. [68]. Copyright 2023, American Chemical Society. (**c**) Schematic illustration of the preparation process of TPU/CNT strain sensor; the relation between GF of TPU/CNTs tube and CNT concentration; the response time of TPU/CNT strain sensor under 1% strain. Reprinted with permission from Ref. [71]. Copyright 2022, MDPI.

**Figure 5 sensors-24-04793-f005:**
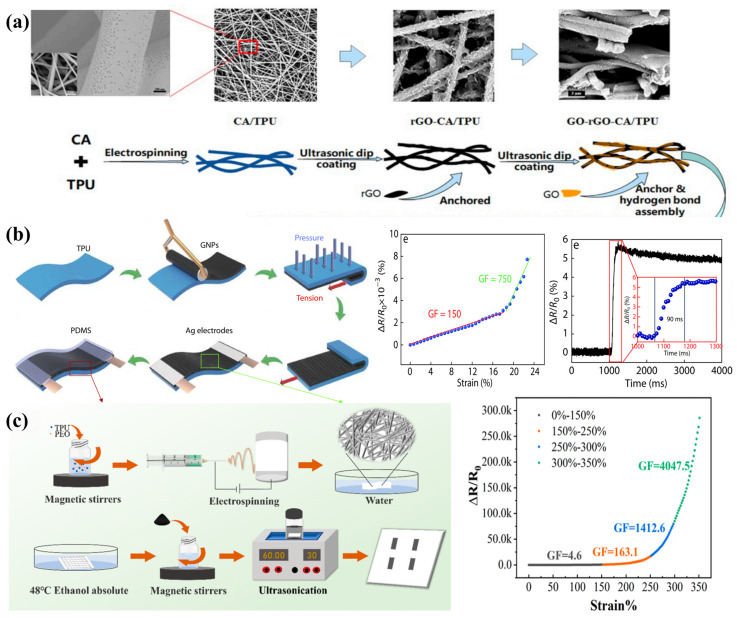
(**a**) Schematic illustration of the whole preparation process of the rGO/GO@CA/TPU strain sensor. Reprinted with permission from Ref. [115]. Copyright 2022, MDPI. (**b**) Schematic illustration of the preparation process of CWS strain sensor; ΔR/R_0_ of CWS strain sensor toward the applied strain; the response time of CWS strain sensor under 0.1% strain. Reprinted with permission from Ref. [73]. Copyright 2020, Springer Nature. (**c**) Schematic illustration of the preparation process of TPU/GNP strain sensor; relative change in resistance and tensile strain curve of TPU/GNP strain sensor. Reprinted with permission from Ref. [74]. Copyright 2024, American Chemical Society.

**Figure 6 sensors-24-04793-f006:**
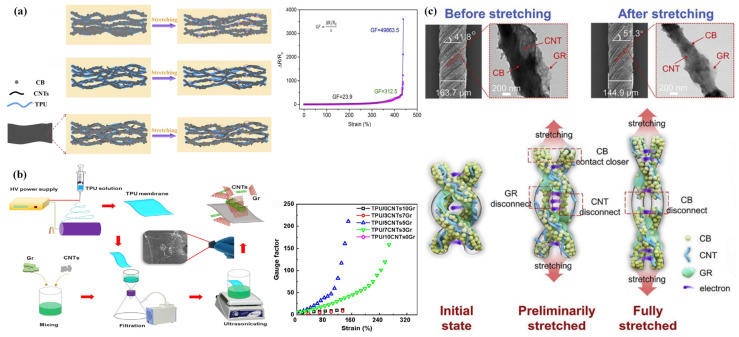
(**a**) Schematic diagram of the strain sensing mechanisms of the CB-TPU, CNT-TPU, (CB+CNTs)-TPU composites with different conductive networks; GF of PDMS/(CB+CNTs)/TPU strain sensor toward the applied strain. Reprinted with permission from Ref. [77]. Copyright 2022, Elsevier. (**b**) Schematic illustration of the preparation process of TPU/CNTs-Gr strain sensor; GF of TPU/CNTs-Gr strain sensor with different composition content toward the applied strain. Reprinted with permission from Ref. [80]. Copyright 2022, Wiley. (**c**) Structural evolution of the ternary conductive network SEM images of 24CB/3CNT/3GR-Y and TEM images of inside nanofiber before stretching and after stretching; schematic illustration of structural evolution of ternary conductive network under stretching. Reprinted with permission from Ref. [81]. Copyright 2022, Elsevier.

**Figure 7 sensors-24-04793-f007:**
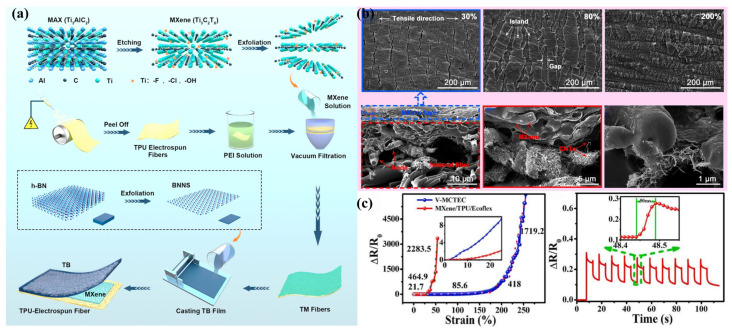
(**a**) Schematic illustration of the whole preparation process of the TPU/MXene/TPU-BNNS strain sensor. Reprinted with permission from Ref. [83]. Copyright 2024, American Chemical Society. (**b**) Crack morphology changes of MXene/CNT/TPU strain sensor under stretching strain of 30%, 80%, and 200%, respectively; cross-section SEM images of MXene/CNT/TPU composite film at different magnifications. Reprinted with permission from Ref. [84]. Copyright 2022, American Chemical Society. (**c**) Resistance changes with the strain of the vertical MXene-CNT/TPU/Ecoflex composites (V-MCTEC) and MXene/TPU/Ecoflex composites; the response time of V-MCTEC strain sensor under 0.1% strain. Reprinted with permission from Ref. [85]. Copyright 2024, Elsevier.

**Figure 8 sensors-24-04793-f008:**
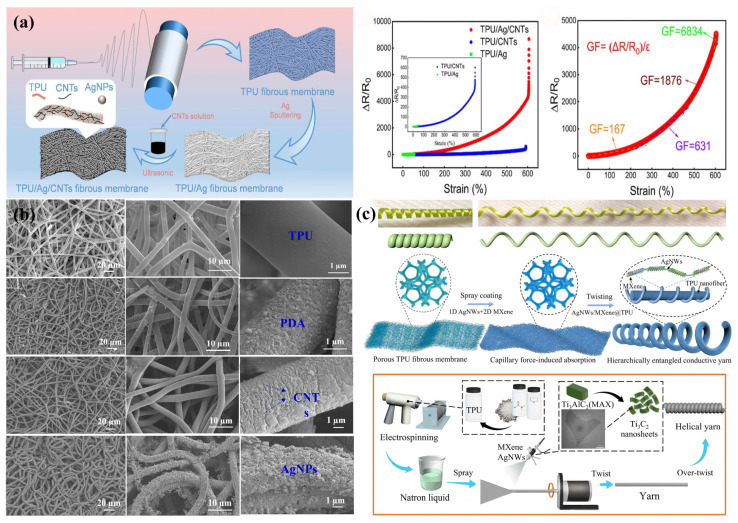
(**a**) Schematic illustration of the whole preparation process of the TPU/Ag/CNT fibrous membrane; relative resistance−strain sensing behaviors’ comparison of the TPU/Ag/CNT strain sensor, TPU/CNT strain sensor, and TPU/Ag strain sensor; sensitivity of the TPU/Ag/CNT strain sensor in four regions from a 0 to 604% strain. Reprinted with permission from Ref. [86]. Copyright 2024, American Chemical Society. (**b**) SEM images of the surface for pure TPU mat (line 1), PDA/TPU mat (line 2), CNT/PDA/TPU mat (line 3), and AgNP/CNT/PDA/TPU composite (line 4), respectively. Reprinted with permission from Ref. [141]. Copyright 2023, Elsevier. (**c**) Photograph and schematic diagram of stretchable curling structure of climbing plant under original and stretching state; the structure evolution of hierarchically interlocked helical conductive yarn; schematic diagram of fabricating hierarchically interlocked helical conductive yarn via overtwisting the AgNWs/MXene multi-dimensional synergistic conductive network that entangled with elastic TPU nanofibers. Reprinted with permission from Ref. [87]. Copyright 2023, Elsevier.

**Figure 9 sensors-24-04793-f009:**
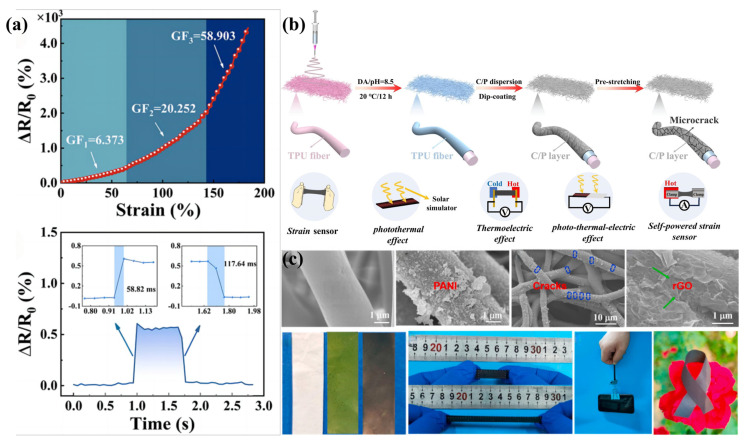
(**a**) GF of SCFY strain sensor toward the applied strain; the response time of SCFY strain sensor under 1% strain. Reprinted with permission from Ref. [90]. Copyright 2023, American Chemical Society. (**b**) Preparation process and characterization of CPT conductive fibrous membrane. Reprinted with permission from Ref. [91]. Copyright 2024, Elsevier. (**c**) SEM images of virgin TPU, PANI/TPU mats, and PANI/TPU mat after pre−stretching and GPTM; digital images of TPU, PANI/TPU, and the GPTM; the stretchability of the GPTM; photograph of GPTM hanging a phone; lightweight and flexibility of samples. Reprinted with permission from Ref. [92]. Copyright 2022, Elsevier.

**Figure 10 sensors-24-04793-f010:**
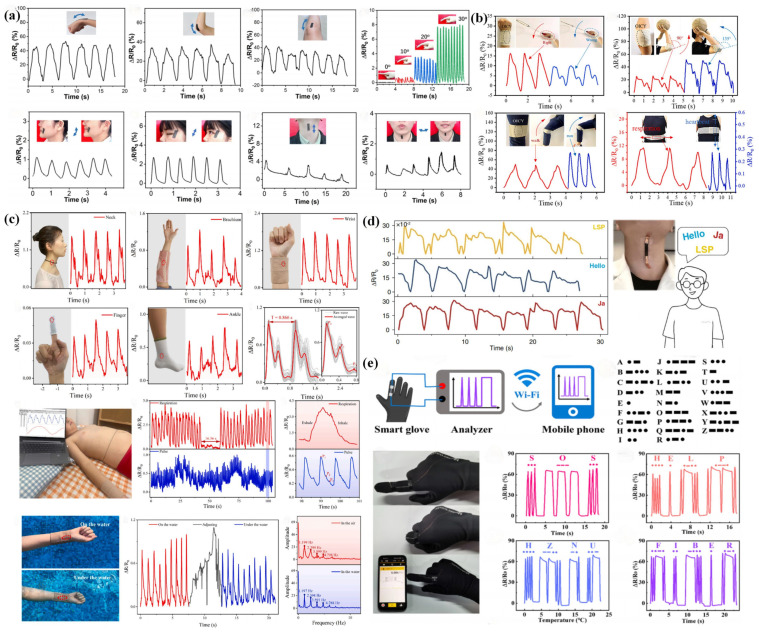
(**a**) Monitoring human body movement via a corresponding relative resistance change (ΔR/R_0_) signal of the CNT/F-TPU strain sensor: wrist movement; elbow movement; knee bending; finger movement with different bending degrees (0−30°); microexpressions of smiling; blinking; swallowing; pronouncing the word “work”. Reprinted with permission from Ref. [68]. Copyright 2023, American Chemical Society. (**b**) Application of the smart sports bandage in badminton, basketball, walking and running, monitoring respiration and heartbeat. Reprinted with permission from Ref. [25]. Copyright 2022, Elsevier. (**c**) Demonstration of the KMYS for physiological signal detection under different conditions: pulse waveform at the carotid artery, brachial artery, radial artery digital artery, and ankle artery; the pulse signals (gray) and averaged pattern (red) using the KMYS; photograph showing the KMYS was worn against the abdomen of an overweight 10-year-old child for respiration monitoring; the respiration and pulse signals from the children during sleeping; the magnified views of respiration and pulse signals; photograph showing the KMYS was attached to the wrist on and under the water, respectively; pulse signals in different states; spectral information embedded within the pulse wave. Reprinted with permission from Ref. [129]. Copyright 2024, Elsevier. (**d**) Sensing application of the TPU/CB strain sensor in human motion: speaking different words with sample fixing at the neck. Reprinted with permission from Ref. [64]. Copyright 2021, Springer. (**e**) Information encryption and transmission based on Morse code realized by the IL/TPU ionogel fiber-shaped strain sensor; a scheme for connecting the sensor to a wireless transmitter; schematic diagram of Morse code; photograph of the sensor attached to a finger to send information; encryption and translation of “SOS”, “HELP”, “HZNU”, and “FIBER”. Reprinted with permission from Ref. [160]. Copyright 2022, American Chemical Society.

**Table 1 sensors-24-04793-t001:** Effect of electrospinning process parameters on morphological properties of TPU.

Process Parameter	Description	Effect on Morphological Properties
Nozzle Axis Type	Single Axis Nozzle	Uniform diameters, limited control over orientation and porosity
	Coaxial Nozzle	Increases nanofiber diameter, more porous structure, enhanced control over morphology
Collector Type	Rotating Cylinder Collector	Larger diameters, less effective in fiber alignment
	Rotational Wire Collector	Thinner, more oriented fibers, better alignment due to smaller contact area and higher rotation speeds
Collector Speed	Low Speed	Random fiber orientations, limited impact on diameter
	High Speed (2000 rpm)	Improved orientation, increased fiber diameter due to relaxation post-spinning, optimal for aligned fiber applications
Filler Types	Inorganic Fillers (e.g., Salt Particles)	Thinnest, highly oriented fibers, enhanced alignment
	Organic Fillers (e.g., Polystyrene)	Modified morphology, impact on mechanical properties, and surface characteristics
Fiber Diameter	Increases with Collector Speed	Larger diameters at higher speeds due to relaxation
	Increases with Coaxial Nozzle	Thicker fibers compared to single-axis nozzles
Fiber Orientation	Enhanced with Higher Rotational Speeds	Speeds up to 2000 rpm needed for significant orientation, critical for specific applications
	Better with Wire Collector	Superior alignment compared to cylindrical collectors
Surface Porous Structure	Coaxial Technique	Highly porous surfaces with low boiling point solvents for core, formation of hollow and porous fibers
	Influenced by Solvent Properties	Control porosity and surface characteristics based on solvent boiling points

**Table 2 sensors-24-04793-t002:** Main features of various flexible strain sensors.

Conductive Material	Materials	Structure	Sensitivity (at Strain %)	Sensing Range (%)	Response/Recovery Time (ms)	Durability (Cycles at Strain %)	Ref
0D CB	CB/TPU	membrane	8962.7 (160)	0–160	60	10,000 (10)	[64]
	CB/TPU/Ecoflex	sandwich	3186.4 (225)	0.5–225	70	5000 (40)	[65]
1D CNTs	DATPU/CNTs	membrane	1200 (710)	0–710	300	15,000 (100)	[66]
	CNTs/DA/TPU	membrane	10,528.53 (200)	0–200	188/211	300 (10)	[67]
	CNTs/F-TPU	mat	2.86 (150)	0–550	110	5000 (30)	[68]
	CNTs/TPU@SBS	double-percolation	32,411 (100)	0.2–100	214	500 (10)	[69]
	CNTs/TPU	membrane	1571 (400)	0–400	-	10,000 (10)	[70]
	TPU/CNTs	tube	57 (760)	2–760	45	10,000 (200)	[71]
2D Gr	PDA/rGO/TPU	mat	6583 (150)	0–150	100	9000 (50)	[72]
	GNPs/TPU/PDMS	crack-wrinkle	750 (24)	0.1–24	90	20,000 (5)	[73]
	TPU/GNPs	porous-convoluted	4047.5 (350)	0–350	-	10,000 (50)	[74]
multidimensionality	PDA/ CB/CNF/TPU	membrane	312.4 (285)	0.12−285	287/457	1000 (50)	[75]
	TPU/CB-CNTs	fiber	6.0 (3.0)	0–280.5	248	2000 (5)	[76]
	PDMS/(CB+CNTs)/TPU	mat	49,863.5 (437.9)	0–437.9	-	1300 (50)	[77]
	CB/ MWCNTs/TPU	membrane	5705.53 (150)	0–150	220	6000 (7)	[78]
	TPU/SCB@CNTs/F-SiO_2_	film	60.42 (100)	0–100	75/100	1000 (70)	[79]
	TPU/CNTs-Gr	membrane	217 (172)	0–172	-	10,000 (30)	[80]
	CB/CNTs/GR/TPU	yarn	17.74 (100)	0–227	220	5000 (50)	[81]
MXene	MXene/PDA/TPU	membrane	2600 (200)	0.1–200	77/186	1000 (50)	[82]
	TPU/MXene/BNNS	sandwich	2080.9 (100)	0–100	-	2500 (50)	[83]
	MXene/CNTs/TPU	bilayer-conductive	2911 (330)	0–330	80	2600 (50)	[84]
	MXene-CNTs/TPU/Ecoflex	sandwich	1719.2 (251)	0.1–251	50	10,000 (50)	[85]
Ag	TPU/Ag/CNTs	membrane	6834 (604)	0.1–604	122/164	1000 (50)	[86]
	AgNWs/MXene/TPU	yarn	1.7 (1000)	0–1000	100	3000 (100)	[87]
LM	LM/TPU	membrane	0.2 (200)	0–200	290/360	9000 (75)	[88]
	TPU/LM/NFs	yarn	6 (200)	0–548	-	1000 (50)	[89]
conductive polymer	PDA/PPy/TPU-TPE	fiber	58.9 (184)	0–184	58.82/117.64	2000 (20)	[90]
	MWCNT/PEDOT:PSS/TPU	membrane	6008.3 (680)	0.5–680	200	6000 (50)	[91]
	rGO/PANI/TPU	mat	3000.2 (300)	0.1–300	90	10,000 (40)	[92]

## Data Availability

The data used to support the findings of this study are available from the corresponding author upon request.

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
