# Peer review of "Flexible Strain Sensors Based on Thermoplastic Polyurethane Fabricated by Electrospinning: A Review"

_sensors, 2024, doi:10.3390/s24154793_

Round 1

Reviewer 1 Report

Comments and Suggestions for Authors

Dear editor,

By reviewing the manuscript "Flexible Strain Sensors Based on Thermoplastic Polyurethane Fabricated by Electrospinning: A Review" (Manuscript ID: sensors-3088765) by Jingjing Zhang. My suggestion is "revised for publication". Our comments are as follows:

In this paper, the working principle of strain sensors and electrospinning technology are analyzed and compared, then the research progress of various conductive fillers based on electrospun TPU flexible strain sensors are reviewed, and then their application fields are discussed in detail. However, there are still some problems in the article, mainly the following points:

1.        What are the advantages of TPU compared to other polymers?

2.        What are the advantages of electrospinning compared to other preparation processes?

3.        Why review the research progress of flexible strain sensor based on electrospun TPU?

4.        It is recommended to unify the format of Table 1.

5.        The authors need to double-check the reference format, the title writing format of the cited literature is not uniform.

6.        No drawings (a), (b), (c) or (d) are indicated in Figure 9.

7.        It is recommended to read and revise the sentences of the article carefully.

8.        Some recent advances and applications of high-performance sensors can be referenced to improve the readability of the article, as:

Nano Energy, 2023, 108, 108215

Comments on the Quality of English Language

 It is recommended to read and revise the sentences of the article carefully.

Author Response

Dear Professor            ,

Thank you very much for your attention to our article “Flexible Strain Sensors Based on Thermoplastic Polyurethane Fabricated by Electrospinning: A Review” (Manuscript Number: sensors-3088765). The comments from referees are very valuable and helpful for improving our article. All the authors have seriously discussed about these comments. We have accordingly revised the manuscript to meet with the requirements of the journal “Sensors”. Our changes are included in the revised manuscript marked by the RED font and the detailed point-by-point responses to the comments are listed as RED font as following.

Comment 1: What are the advantages of TPU compared to other polymers?

Response 1: Thank you for your suggestion regarding the issues in our manuscript. Regarding the advantages of TPU over other polymers, we emphasized this on page 2, lines 52-67 of the revised draft.

Comment 2: What are the advantages of electrospinning compared to other preparation processes?

Response 2: In response to the reviewer's question "Advantages of electrospinning over other preparative processes", we have added a comparison between electrospinning and other preparative processes in lines 159-170 on page 4 of the revised manuscript to highlight the unique advantages of electrospinning.

Comment 3: Why review the research progress of flexible strain sensor based on electrospun TPU?

Response 3: To address the lack of a fully motivated explanation, we emphasize the unique focus and novelty of our work in lines 68-92 on page 3 of the revised draft. Over recent years, many research results have been achieved in the study of electrospun TPU flexible strain sensors. However, there is a lack of review papers that provide a systematic and comprehensive summary and outlook of these recent research results. This review comprehensively and systematically summarizes the recent advances in electrospun TPU-based flexible strain sensors, bridges the gap in the field, and serves as a unique guideline for subsequent research on electrospun TPU-based flexible strain sensors.

Comment 4: It is recommended to unify the format of Table 1.

Response 4: In response to the reviewer's suggestions for Table 1, we revised the format of Table 1 and included additional papers and data in Table 1.

Comment 5: The authors need to double-check the reference format, the title writing format of the cited literature is not uniform.

Response 5: In response to the reviewer's suggestions on the formatting of references, we double-checked and corrected the formatting of title writing for all cited documents.

Comment 6: No drawings (a), (b), (c) or (d) are indicated in Figure 9.

Response 6: Thanks to the reviewer's careful review, we have added the missing labeling in Figure 9.

Comment 7: It is recommended to read and revise the sentences of the article carefully.

Response 7: Thank you for your suggestion regarding the issues in our manuscript. We have carefully read and revised the English expression of the sentences in the article.

Comment 8: Some recent advances and applications of high-performance sensors can be referenced to improve the readability of the article, as: Nano Energy, 2023, 108, 108215

Response 8: Thank you for your suggestion regarding the issues in our manuscript. We decided to follow your suggestion to add recent advances and applications of high performance sensors including the paper you mentioned to improve the readability of the review.

Again, many thanks for your editorial endeavors on behalf of all of the authors and we do appreciate very much the constructive comments and good suggestions from the reviewers such that we were able to improve the overall quality and clarity of our paper. Hopefully, we could have our article been considered of publication in your journal. We really believe this work would be sufficient novelty and impact to appeal to your readership. Should there been any other corrections we could make, please feel free to contact us.

Sincerely Yours,

Jingjing Zhang

Tropical Agriculture and Forestry

Hainan University

Danzhou 571737, China

E-mail: 993958@hainu.edu.cn.

Reviewer 2 Report

Comments and Suggestions for Authors

This work reviewed recent progress of electrospun TPU flexible strain sensors with a focus on the incorporation of various conductive fillers such as carbonaceous materials, MXene, metallic materials, and conductive polymers. The applications of  TPU flexible strain sensors were also discussed. Strain sensors are very interesting and important topics for materials science. However, I don't think the review is properly organized to be published in Sensors in the current version. A major revision should be made based on the following reasons.

1. The Abstract is too weak to attract readers in this area, and no any essential challenges/limitations were put forward to show why this Review is of important significance.

2. The introduction section is not well organized. More references should be cited related to electrospun TPU strain sensors. And compared with current reviews of strain sensors, the innovation of this review should be addressed. The original contributions of the authors should be highlighted.

3. Section 2.2. Electrospinning Techniques, only 2 references were cited. Additionally, the advantages of the electrospinning method compared to other preparation methods are not listed.

4. The article provides an extensive introduction of The Use of Conductive Fillers in Electrospun TPU Strain Sensors. For different fillers, it's best to include a brief summary at the end of each part.

5. For the section of applications, subtitles can be added to make the content look more organized.

6. The authors didn't well conclude the theoretical foundations and future developments of the strain sensors. More critical points should been given.

Comments on the Quality of English Language

The English writing should be improved, and typo/grammar errors need to correct properly. For example in Abstract: sen-sors, var-ious. 

Author Response

Dear Professor            ,

Thank you very much for your attention to our article “Flexible Strain Sensors Based on Thermoplastic Polyurethane Fabricated by Electrospinning: A Review” (Manuscript Number: sensors-3088765). The comments from referees are very valuable and helpful for improving our article. All the authors have seriously discussed about these comments. We have accordingly revised the manuscript to meet with the requirements of the journal “Sensors”. Our changes are included in the revised manuscript marked by the RED font and the detailed point-by-point responses to the comments are listed as RED font as following.

Comment 1: The Abstract is too weak to attract readers in this area, and no any essential challenges/limitations were put forward to show why this Review is of important significance.

Response 1: Thank you for your suggestion regarding the issues in our manuscript. We have decided to follow your suggestion to revise the abstract section of the paper to emphasize the unique importance of this review and to increase the reader's interest in reading it.

Comment 2: The introduction section is not well organized. More references should be cited related to electrospun TPU strain sensors. And compared with current reviews of strain sensors, the innovation of this review should be addressed. The original contributions of the authors should be highlighted.

Response 2: In response to the reviewer's comment that "the introduction is not well organized", we emphasize the unique focus and novelty of our work in lines 68-92 on page 3 of the revised draft. Over recent years, many research results have been achieved in the study of electrospun TPU flexible strain sensors. However, there is a lack of review papers that provide a systematic and comprehensive summary and outlook of these recent research results. This review comprehensively and systematically summarizes the recent advances in electrospun TPU-based flexible strain sensors, bridges the gap in the field, and serves as a unique guideline for subsequent research on electrospun TPU-based flexible strain sensors.

Comment 3: Section 2.2. Electrospinning Techniques, only 2 references were cited. Additionally, the advantages of the electrospinning method compared to other preparation methods are not listed.

Response 3: In response to the reviewer's suggestion of "Section 2.2. Electrospinning Techniques", we have cited more references related to electrospinning technology, and added a comparison between electrospinning and other preparative processes in lines 138-170 on page 4 of the revised manuscript to highlight the unique advantages of electrospinning.

Comment 4: The article provides an extensive introduction of “The Use of Conductive Fillers in Electrospun TPU Strain Sensors”. For different fillers, it's best to include a brief summary at the end of each part.

Response 4: In response to the reviewer's suggestions for each part of conductive fillers, we decided to adopt and add a brief summary at the end of each part.

Comment 5: For the section of applications, subtitles can be added to make the content look more organized.

Response 5: In response to the reviewer's suggestions on the section of applications, we split the entire app section into three subsections and added subheadings to make the app content look more organized.

Comment 6: The authors didn't well conclude the theoretical foundations and future developments of the strain sensors. More critical points should been given.

Response 6: Thank you for your suggestion regarding the issues in our manuscript. We have revised the "Conclusions and Outlook" section and given more key points to better summarize the theoretical basis and future development of strain sensors.

Comment 7: The English writing should be improved, and typo/grammar errors need to correct properly. For example in Abstract: sen-sors, var-ious.

Response 7: Thank you for your suggestion regarding the issues in our manuscript. We have carefully read and revised the English expression of the sentences in the article.

Again, many thanks for your editorial endeavors on behalf of all of the authors and we do appreciate very much the constructive comments and good suggestions from the reviewers such that we were able to improve the overall quality and clarity of our paper. Hopefully, we could have our article been considered of publication in your journal. We really believe this work would be sufficient novelty and impact to appeal to your readership. Should there been any other corrections we could make, please feel free to contact us.

Sincerely Yours,

Jingjing Zhang

Tropical Agriculture and Forestry

Hainan University

Danzhou 571737, China

E-mail: 993958@hainu.edu.cn.

Reviewer 3 Report

Comments and Suggestions for Authors

Zhiyuan Zhou et al. have written a comprehensive review on strain sensors based on TPU manufactured via electrospinning. While the paper is generally well-written, several issues need to be addressed before it can be published:

  1. More data need to be added to Table 1, including hysteresis, transient response (response and recovery time), maximum elongation, and linearity.
  2. Additional papers should be incorporated into Table 1.
  3. A table, figure, or discussion on the influence of electrospinning on performance parameters should be included.
  4. Regarding the statement in the introduction: "With the rapid advances in the Internet of Things (IoT), Industry 4.0, big data, artificial intelligence (AI), robotics, and digital health, there is an increasing demand for high-precision sensing at all places and times," please introduce practical examples of sensors in these applications in the final sections.
  5. A section of "future trends" should be incluyed.
Comments on the Quality of English Language

Minor editing of English language required.

Author Response

Dear Professor            ,

Thank you very much for your attention to our article “Flexible Strain Sensors Based on Thermoplastic Polyurethane Fabricated by Electrospinning: A Review” (Manuscript Number: sensors-3088765). The comments from referees are very valuable and helpful for improving our article. All the authors have seriously discussed about these comments. We have accordingly revised the manuscript to meet with the requirements of the journal “Sensors”. Our changes are included in the revised manuscript marked by the RED font and the detailed point-by-point responses to the comments are listed as RED font as following.

Comment 1: More data need to be added to Table 1, including hysteresis, transient response (response and recovery time), maximum elongation, and linearity.

Response 1: Thank you for your suggestion regarding the issues in our manuscript. We have added two new columns in Table 2 (Original Table 1): "response/recovery time" and "durability" to add more data.

Comment 2: Additional papers should be incorporated into Table 1.

Response 2: Thank you for your suggestion regarding the issues in our manuscript. We have cited more references in Table 2 (Original Table 1).

Comment 3: A table, figure, or discussion on the influence of electrospinning on performance parameters should be included.

Response 3: Thank you for your suggestion regarding the issues in our manuscript. We introduced Table 1 to demonstrate the effect of electrospinning process parameters on the morphological properties of TPU.

Comment 4: Regarding the statement in the introduction: "With the rapid advances in the Internet of Things (IoT), Industry 4.0, big data, artificial intelligence (AI), robotics, and digital health, there is an increasing demand for high-precision sensing at all places and times," please introduce practical examples of sensors in these applications in the final sections.

Response 4: In response to the reviewer's suggestions on the section of applications, we have presented practical examples of sensors in applications such as the Internet of Things (IoT), Industry 4.0, Big Data, Artificial Intelligence (AI), Robotics and Digital Health in our Applications section.

Comment 5: A section of "future trends" should be incluyed.

Response 5: Thank you for your suggestion regarding the issues in our manuscript. We have merged the section on "future trends" into the section on "Conclusions and Outlook" and given more key points to better summarize the theoretical basis and future development of flexible strain sensors.

Comment 6: Minor editing of English language required.

Response 6: Thank you for your suggestion regarding the issues in our manuscript. We have carefully read and revised the English expression of the sentences in the article.

Again, many thanks for your editorial endeavors on behalf of all of the authors and we do appreciate very much the constructive comments and good suggestions from the reviewers such that we were able to improve the overall quality and clarity of our paper. Hopefully, we could have our article been considered of publication in your journal. We really believe this work would be sufficient novelty and impact to appeal to your readership. Should there been any other corrections we could make, please feel free to contact us.

Sincerely Yours,

Jingjing Zhang

Tropical Agriculture and Forestry

Hainan University

Danzhou 571737, China

E-mail: 993958@hainu.edu.cn.

Round 2

Reviewer 1 Report

Comments and Suggestions for Authors

Dear editor,

In this paper, the working principle of strain sensors and electrospinning technology are analyzed and compared, then the research progress of various conductive fillers based on electrospun TPU flexible strain sensors are reviewed, and then their application fields are discussed in detail. However, there is still a problem in the article:

1. The authors need to double-check the reference format, the title writing format of the cited literature is not uniform.

Comments on the Quality of English Language

 Minor editing of English language required

Reviewer 2 Report

Comments and Suggestions for Authors

The revised manuscript could be accepted for publication in Sensors.

Reviewer 3 Report

Comments and Suggestions for Authors

The authors have considered all the feedback provided in the comments.